# Effect of the Active Aging-in-Place–Rehabilitation Nursing Program: A Randomized Controlled Trial

**DOI:** 10.3390/healthcare11020276

**Published:** 2023-01-16

**Authors:** Ana da Conceição Alves Faria, Maria Manuela F. P. S. Martins, Olga Maria Pimenta Lopes Ribeiro, João Miguel Almeida Ventura-Silva, Esmeralda Faria Fonseca, Luciano José Moreira Ferreira, José Alberto Laredo-Aguilera

**Affiliations:** 1Abel Salazar Biomedical Sciences Institute, University of Porto, 4050-313 Porto, Portugal; 2Aces Ave/Famalicão, North Region Health Administration, 4000-447 Porto, Portugal; 3Nursing School of Porto (ESEP), 4200-072 Porto, Portugal; 4CINTESIS@RISE, 4200-450 Porto, Portugal; 5Centro Hospitalar Universitário de São João, 4200-319 Porto, Portugal; 6Facultad de Fisioterapia y Enfermería, Campus de Fábrica de Armas, Universidad de Castilla-La Mancha, 45071 Toledo, Spain; 7Multidisciplinary Research Group in Care (IMCU), University of Castilla-La Mancha, 45005 Toledo, Spain

**Keywords:** aged, frailty, lifestyle, physical exercise, rehabilitation, nursing

## Abstract

(1) Background: It is of great importance to promote functional capacity and positive lifestyles, since they contribute to preventing the progression of frailty among the older adults. The aim of this study was to evaluate the effect of active aging-in-place–rehabilitation nursing program (AAP-RNP) on the functional capacity and lifestyles of frail older adults. (2) Methods: This was a single-blinded, two-group, randomized, controlled trial of 30 frail older people enrolled at a Health-care unit in Portugal between 2021 and 2022. The duration of the program was 12 weeks, and the sessions took place at the participants’ homes. We used as instruments the Tilburg Frailty Indicator; Fried frailty phenotype; Senior Fitness Test battery; Barthel Index; Lawton Index; handgrip strength measurement; Tinetti Index; Individual lifestyle profile; and Borg’s perception of effort. (3) Results: Post-program, there was an improvement in multidimensional and physical frailty, functional capacity, balance, and perceived exertion (*p* < 0.05) in the experimental group. Among the older adults’ lifestyles, we observed significant improvements in physical activity habits, relational behavior, and stress management. (4) Conclusions: Rehabilitation nurses have a relevant role, and the AAP-RNP seems to be effective in improving functional capacity and lifestyles in frail older adults.

## 1. Introduction

An increase in the older population in Portugal, Europe, and throughout the world, especially in developed countries, has led to a higher prevalence of frailty, dependency, multimorbidities, and disabling disease among older adults [1]. The worldwide prevalence of frailty is 4 to 16% of older adult individuals over the age of 65, and 25% of those aged 85 and older [2]. Aging does not inevitably lead to frailty; however, with advancing age, these processes are more prevalent and, hence, they require individuals to have a greater network of health and social care [2].

The literature review on frailty approaches different models and concepts. Some of these focus exclusively on the physical domain, as proposed by Fried, who defines frailty in terms of phenotypic markers such as global weakness with low muscle strength, decreased mobility and general slowness, especially concerning walking, fatigue, or exhaustion, decreased physical activity, and unintentional weight loss [3]. Another definition considers frailty as an age-related condition characterized by a weakened response to stressful events associated with numerous pathophysiological modifications in different body systems, increasing the risk development and appearance of chronic and disabling diseases, geriatric syndromes, and disabilities [4]. In addition to the cognitive and physical components, other authors have also added social components. They, therefore, define frailty as a dynamic condition that affects the person experiencing damage in their psychological (including aspects related to cognition and mood), physical, and/or social domains combined with the effects of life course determinants. Numerous variables can cause these conditions and increase the risk of adverse health-related outcomes [5].

Among other consequences, frail older adults limit their daily activities, decrease physical activity, live a sedentary lifestyle, decrease their social participation [6,7,8], and often take refuge at home, either because of their fear of falling or due to greater locomotor and cognitive difficulties and their stigma, thus intensifying the aggravation of the process and the older adults’ quality of life [9,10].

Despite the lack of consensus on the definition, screening methods and assessment of frailty reveal that early identification and intervention can significantly reduce frailty and functional dependence in older people and contribute to maintaining physical, cognitive, and social abilities for as long as possible [11].

Among the non-pharmacological interventions to treat, prevent, and delay frailty, we consider exercise the most effective strategy regardless of how frailty is measured [12,13,14,15,16,17]. However, although we can find different resources in the community such as group programs to promote active aging, sometimes people with frailty signs do not adhere to these programs. Some common barriers cited by older adults to these programs include health-related factors (chronic illness, functional disability, pain/discomfort, fear of falling, or thinking they are unable to be physically active) [18], environmental factors (access difficulties or lack of transportation) [19], lack of interest and motivation for physical activity, lack of knowledge and doubts about the social and health benefits of exercise [20], and time constraints (the amount of time needed to commute and the time to perform exercise) [21].

For many older persons, conventional physical exercise is not attractive [22], and in several studies, older people have reported a preference for being physically active through activities they enjoy, such as walking and gardening [22,23] or individual approaches at home aimed at maintaining physical function [24]. To this end, structured and individualized multicomponent programs that combine strength training, endurance, balance, flexibility, and functional tasks must be a strategy for improving older adults’ functional capacity and independence in the activities of daily living (ADL) [13,15,22,23,24,25].

In addition to the importance of individualization, studies suggest that to improve program adherence and overcome decreased physical activity and relational behavior among frail older adults, it is necessary to include program strategies which promote behavioral change [21,22,23] and base their design on the health action process approach model [26] and Nola Pender’s Health Promotion Model [27]. These focus on increasing motivation and analyzing barriers, situations, and feelings that hinder behavioral adherence. With the support and aid of a health professional who promotes individualized physical exercise adapted to older adults’ specific needs and difficulties [22,23,24], makes them aware of the importance of physical activity, and motivates them to participate and interact socially in a group [28,29], it is possible to delay frailty while aging in place and avoiding institutionalization [30]. 

However, while personalized, home-based, multicomponent exercise programs could be a valuable solution for frail older people, the evidence on their effect is still scarce. Previous studies of individualized multicomponent exercises at home for frail older people consisted of interventions with limited health professional supervision, and no studies are known to assess the impact of these programs on older adults’ lifestyles. Regarding functional capacity, the results of previous individualized, home-based studies are scarce, and we are not aware of any study in which professionals’ interventions are in the context of frail older adults; thus, the physical, architectural, personal, or social barriers that limit adherence to physical exercise are unknown [12,13,14,15,16,17].

Thus, in an attempt for health care to meet older adults’ individual needs, and since the evidence of the effectiveness of supervised, multicomponent, home-based programs is still scarce, we have recently developed an individualized and structured program. This program is based on mild- to moderate-intensity physical exercise and is called the “Active Aging-in-Place–Rehabilitation Nursing Program” (AAP-RNP). Given the above, the aim of this study was to evaluate the effect of the AAP-RNP on the functional capacity and lifestyle profiles of frail older adults.

## 2. Materials and Methods

### 2.1. Study Design, Setting and Sample

This study was an interventional, single-blind, randomized controlled trial (RCT) with two parallel groups, the experimental group (EG) and the control group (CG), and followed the Consolidated Standards of Reporting Trials statement (CONSORT) [31]. 

The participants included 30 frail older adults enrolled at a health care unit in northern Portugal and recruited by convenience sampling. All had already participated in a previous study with 300 older adults where the frailty profile of the older people was identified, [32] and, after the researcher contacted the older adults by telephone and explained the scope of the program, which took place during October 2021 and January 2022, they agreed to participate in the program. They all met the established inclusion criteria, namely, aged 65 years or older; presenting frailty (assessed using the Tilburg Frailty Indicator (TFI) [33]; lacking any cognitive deficits that would compromise their understanding of the program; having the ability to walk with or without a walking aid; exhibiting mild to moderate dependence assessed using the Barthel Index [34]; having no contraindications to physical exercise; not being institutionalized; being enrolled at a health care unit in northern Portugal; and agreeing to be part of the established rehabilitation program. To ascertain cognitive ability, we evaluated the older adults at the beginning of the interview according to the following items: orientation; memory; volition; and availability. To avoid bias and contamination, the participants were randomly assigned, 15 to the experimental group (EG) and 15 to the control group (CG), via a coin toss (1:1) by an independent researcher who was not involved in the conception and assessment of the program. 

After randomization, the principal researcher, who was not blinded to group allocation, informed each older adult as to which group he or she was allocated. However, to avoid assessment bias, the outcome assessors and data analysts were blinded to the group allocations. Details of the allocation process are shown in Figure 1.

### 2.2. Evaluation and Variables

Both groups were evaluated at two different times (at baseline and after the 12-week program) at the participants’ homes by one independent researcher (a rehabilitation nurse not involved in the conception and implementation program). The EG and GC evaluations started and ended at the same time. An evaluation was carried out at the baseline and post-program for both the EG and CG, always in a 1:1 ratio.

The baseline and post-program forms included sociodemographic characterizations, evaluation of health/disease status, and the application of some instruments described below:Frailty: We assessed multidimensional frailty using the TFI [33]. The TFI is a questionnaire that integrates 3 components (physical, psychological, and social) and is divided into 2 parts: the first part, where the determinants of frailty are recorded, and the second, which is composed of 15 questions divided into 3 components. All items in the second part are rated between 0 and 1, and the cut-off score of frailty was 6. We assessed physical frailty using Fried’s phenotype [35]. We can consider an individual frail if he/she has three or more of the criteria defined in Fried’s phenotype, i.e., 0 points, no frailty; 1 to 2 points, pre-frailty; and 3 to 5 points, frail.Functional Capacity: We used several tests to determine older adults’ functional capacity. The Barthel Index is a questionnaire that assesses dependence in basic activities of daily living (BADL) [34]. It is composed of 10 questions, and the higher the score, which is a maximum of 100 points, the greater the functional independence of the older adult. The Lawton Index assesses dependence on instrumental activities of daily living (IADL) and consists of 8 questions [36]. The score ranges from 0 to 16, and the cut-off points are: 0 to 5, severe or total dependence; 6 to 11, moderate dependence; and 12 to 16, mild dependence or independence. Functional fitness was assessed using Rikli and Jones’ Senior Fitness Test (SFT), a validated test battery for older adults that incorporates the following tests: the chair stand test, counting the number of repetitions in 30 s; the arm-curl test with dumbbells, counting the number of repetitions in 30 s; the back-scratch test, measured in cm; the chair sit-and-reach test, measured in cm; and the timed up-and-go test, recording the time in seconds [37]. For the first five SFT tests, a higher value indicates better performance, while for the last test, lower values indicate better performance. We measured handgrip strength, a predictor of functional capacity in older adults and incorporated in Fried’s phenotype, using a universal hydraulic dynamometer in the dominant hand, choosing the best result from three trials.Balance: We evaluated this parameter using the unipedal balance test and Tinetti’s Index [38]. The Tinetti Index is composed of 16 items, 9 for balance and 7 for gait. The scores range from 0 to 28 points, with low values being associated with decreased balance capacity and an increased risk of falling. A score > 24 points indicates a low fall risk; a score of 19 to 24 points indicates a moderate fall risk; and a score < 19 indicates a high fall risk.Subjective perception of effort was assessed using the Borg scale, which ranges from 0 to 10, with 10 representing maximal effort [39].Lifestyle: We assessed the lifestyle profiles of older people using the Individual Lifestyle Profile (ILP) scale, which is composed of 15 questions subdivided into 5 components, namely: nutrition; physical activity; preventive behavior; relational behavior; and stress control [40]. For each component, the interpretation follows the same logic, but it is suggested to classify the sum of the three questions in each component as follows: up to 3, negative profile; 4 to 6, intermediate (can improve); and 7 to 9, positive profile. The lower the score, the greater the need for behavioral change.

### 2.3. Ethical Procedures

The ethics committee approved the study where the older adults were enrolled (technical advice n. º 24/2020). Participation in the program was voluntary, and all participants signed informed consent forms, preceded by information about the scope of the study and assurance of data confidentiality.

### 2.4. Intervention

Before we defined the structured program, several steps had to be taken. The first was a descriptive, correlational, cross-sectional study that analyzed the frailty profile of the older adults living at home and in the region where we implemented the program. Subsequently, based on these data and an integrative literature review on individualized home exercise programs for frail older adults, namely the components that make up these programs, we constructed the AAP-RNP, which achieved content validity through two focus groups in May 2021. The last step corresponded to the implementation of the AAP-RNP developed by the principal investigator. The program was developed over 12 weeks and included 24 individual sessions (2 sessions per week), each lasting about 60 min.

The planning of the sessions is shown in Table 1.

Before starting the program, we taught the participants safety measures, identification of signs of strain, pain, and each participant’s limits, intensity self-control, and the importance of hydration.

The exercises in each session were adapted according to the ability, needs, and current health status of each older person in terms of intensity, complexity, speed, and time (including dual-task exercises). We designed different strategies to empower older adults to participate actively in the program, to set goals together, to involve them in deciding on the personalized workout plan, and to promote motivation and adherence to physical activity over time both at home and during in-group participation, essential for relational behaviors, with the integration of existing community resources after the 24 sessions. 

The EG received the AAP-RNP, while the CG continued to receive usual health care.

### 2.5. Statistical Analysis

For data processing, we used statistical treatment through the SPSS Statistics program, version 27. We used descriptive statistics to characterize the sample and calculate the mean values of each instrument, taking into account the two moments of assessment and inferential statistics to compare the values of the scales at both moments (baseline and post-program). In the statistical tests, we considered 95% confidence intervals and a significance level of *p* < 0.05. 

For the comparison of the mean values obtained in the initial and final evaluation, we used the non-parametric Wilcoxon test. The significance level of the statistical tests was set at *p* < 0.05.

## 3. Results

As displayed in Table 2, the 30 older adults who participated in this study had a mean age of 81 ± 4.89, ranging from 71 to 86 years. The majority of the older adults were female (70%), widowed (63.33%), and had not completed elementary school (2.48 ± 1.58). Households were composed of a mean of 3 ± 1.12 people. Regarding health status/illness, we found that, on mean, the older adults had 4.41 ± 1.74 diseases and consumed 6.29 ± 1.94 medications.

Most were hypertensive (76.67%) and had problems in their daily lives associated with decreased vision (93.3%) and hearing (70%). Before the program began, the EG and CG were equivalent with respect to gender, age, associated diseases, and frailty status. 

All EG participants had an adherence rate for the 24 sessions of the program greater than 95% (not 100% due to acute illness), with a significant benefit in relation to usual care being observed between the two evaluation moments (baseline and post-program), and there were no differences according to adherence. 

Improvements in statistical significance (*p* < 0.05) in the multidimensional and physical frailty variables assessed using TFI and Fried’s phenotype are shown in Table 3.

Improvements were also observed in the EG in functional fitness, handgrip strength, independence in BADL, balance, and subjective perception of effort. There was an improvement in independence in IADL, however it was not statistically significant, according to Table 4.

After analyzing the results of all domains of the ILP scale, we found that the variables where the most positive change occurred were in relational behavior, physical activity habits, and stress management. There was an improvement in the adherence to adequate nutritional habits, but it was not statistically significant; regarding preventive behavior habits, there were no changes in the EG, and in the CG, the changes were not significant, according to Table 5.

## 4. Discussion

The sociodemographic characteristics of the older adult participants are in line with recent intervention programs in the community with frail older people [43,44,45,46]. There was a predominance of female gender in both groups, as in most published studies [43,44]. The mean age of both EG and CG is in agreement with other interventional programs, which indicates that the age of the older adults does not in itself contraindicate the practice of physical exercise; on the contrary, it demonstrates that it is possible to decrease frailty even among the older adults with advanced age [43,47,48]. Most had low schooling/education, and their marital status was widow(er) in both the EG and the CG as described in studies that analyzed the different factors associated with frailty in older adults [11,49].

Regarding pathological antecedents, we found that most of the older adults reported having multimorbidities and being polymedicated, which is in accordance with several published studies [11,50,51,52,53] and with Rockwood’s model of frailty in relation to the accumulation of deficits [4].

With regard to the effectiveness of the AAP-RNP, significant improvements were observed in both frailty, functional capacity, and lifestyle profiles of the older adults in the EG. A recent study describes a significant association between regular exercise frequency and maintenance or improvement of multidimensional frailty among the older people in the European community over the age of 70 [54]. However, there are no studies that report on individualized programs for frail older adults at home or that assess multidimensional frailty using the TFI.

In 2021, a study was published that involved frail older people in Finland. The authors assessed physical frailty by applying Fried’s phenotype assessment components and concluded that, after implementing individualized exercise programs in older adults’ homes, over a period of 12 months, there were improvements in physical performance and a decrease in the number of falls. However, the program did not prevent the deterioration of dependence in ADLs, nor did it prevent the decrease in handgrip strength [25]. In the present study, we found improvements in the assessment of physical frailty through examining Fried’s criteria with statistical significance, as found in other studies with multicomponent exercise training in which the frailty trajectory did not progress; instead, it decreased after program implementation [43,44].

Regarding functional capacity, we observed significant improvements in the dependence on BADL; however, there were no changes in the assessment of dependence on IADL. To this end, not only were strength, balance, flexibility, and resistance training important, but also functional exercises that, through complex movements, simulated the execution of ADLs. We also observed improvements in functional capacity that were assessed using Barthel and Lawton’s indices in a recent study that incorporated a supervised multicomponent exercise program 5 days a week during 24 weeks [55]. Cadore et al. (2019) identified that multicomponent training not only improves markers of physical frailty but also maintains functional capacity longer throughout aging [56].

In the application of the SFT [37], there was statistically significant improvement in all functional fitness parameters. The muscle strength of the upper limbs was evaluated using the “arm-curl test” with dumbbells at 30 s, and the handgrip strength was evaluated using a dynamometer, and we verified statistically significant changes in both after the intervention program among the older adults of the EG. Several recent studies have also observed improvements in upper limb strength in frail older people after an intervention program. Alvaro Casas-Herrero et al. (2022), through a multicomponent exercise program with seniors that included resistance, balance, flexibility exercises (3 days/week), and gait training (5 days/week) performed for three consecutive months, observed improvements in handgrip strength [57], suggested as a biomarker of aging [58,59] and health status [60,61]. Haider et al. (2017), with the implementation of a strength exercise program in older adults’ homes performed by caregivers after physical therapist training, found improvements in older adults’ handgrip strength and in their physical performance [62].

Regarding the muscle strength of the lower limbs obtained when applying the “chair stand test” for 30 s, a significant improvement was observed in the EG. In the CG, there was a decrease in the number of repetitions between the two moments of evaluation. These results are in agreement with a recent study that reports that a training program that incorporates circuit strength exercises promotes improvements in muscle strength in older adults [63]. Lai et al. (2021) advocates that regular resistance exercise can improve physical fitness in pre-frail older adults [64].

Regarding the upper and lower flexibility evaluated through the “back scratch” test and the “chair sit-and-reach” test, respectively, there was a reduction in the mean distance, with statistical significance. Previous studies with older people in the community have proven that multicomponent training twice or more per week improves flexibility among the older adults [65,66].

Concerning balance assessed using the SFT “single-leg balance” test, Tinetti’s Index, and “timed up-and-go” test, improvements were observed in the EG in all parameters. A recent meta-analysis demonstrates that multicomponent exercise can improve balance and muscle strength in frail older people, and endurance improves significantly, as the intervention lasts longer than 12 weeks [67]. Recent research reports that physical activity has an impact on the physical performance of frail older adults, namely balance and gait speed, with implications for fewer falls and older adults’ quality of life [62,68]. Perez-Sousa et al. (2019) report that aging is associated with a greater decline in the lower body than in the upper body, and these changes may be a cause of the decline in gait speed and instability [69]. However, despite the improvements observed in our study, it will be necessary to continue balance training, given that these older people maintain a high risk of falling, either through evaluation of the Tinetti Index or through evaluation of the “timed up-and-go” test.

With regard to older adults’ lifestyles profiles, we found that both in the EG and in the CG, older people had negative lifestyles profiles, and when analyzing the five components of the ILP scale, we found that the worst results were associated with physical activity and relational behavior [40]. After the implementation of this program, we verified that older adults were more physically active with less fear of falling, and became more participative at the social level, establishing relational behaviors with other older people. These findings meet those described in several studies that show that improved functional capacity promoted through physical activity, even in simple movements at home when performing BADL, improves social participation and relational behavior with reflections on the health and well-being of older adults [70,71]. Tarazona-Santabalbina et al. (2016), whilst implementing an individualized multicomponent exercise program with frail older adults, not only observed improvements in physical and cognitive function, but also in emotional and social networks in frail older adults [55].

The main strength of our study was that the program was implemented in the context of where the older adults live. Thus, it was possible to observe the existing barriers at home and to perceive the difficulties that older adults have that limit them in the practice of physical exercise. With this individualized program that met the needs and difficulties of frail older people, and with the implementation of strategies that motivated them to change their behavior, it was possible to verify improvements in the functional capacity and lifestyles of frail older people.

As a limitation of the present study, we can mention the fact that the sample was small (conditioned by COVID-19 pandemic restrictions) and it only included one of Portugal’s regions. In the future, it will be relevant to implement this program in another context, with a larger sample and with more rehabilitation nurses. 

## 5. Conclusions

The results of this study provide evidence on the fact that supervised, individualized, and progressive multicomponent exercise programs performed at older adults’ homes and focused on motivation and engagement for behavioral change, with twice weekly sessions during short periods of time and low cost intervention, improve frailty, functional capacity, and older adults’ lifestyles.

Thus, we can conclude that individualized home exercises are feasible and beneficial for frail older adults living at home and should be included and implemented by nurses in the community as primary agents in promoting health and healthy lifestyles.

Although group programs are already being developed in the community with this purpose and with benefits in relational behaviors, sometimes older people report difficulty in access, mobility, and fear of participating in groups; therefore, individualized strategies should be developed according to the needs of each older person which will thus improve their adherence. Family health teams must improve the advice they give to users concerning these programs, as well as the allocation of rehabilitation nurses to these projects so that older people can have a better quality of life.

## Figures and Tables

**Figure 1 healthcare-11-00276-f001:**
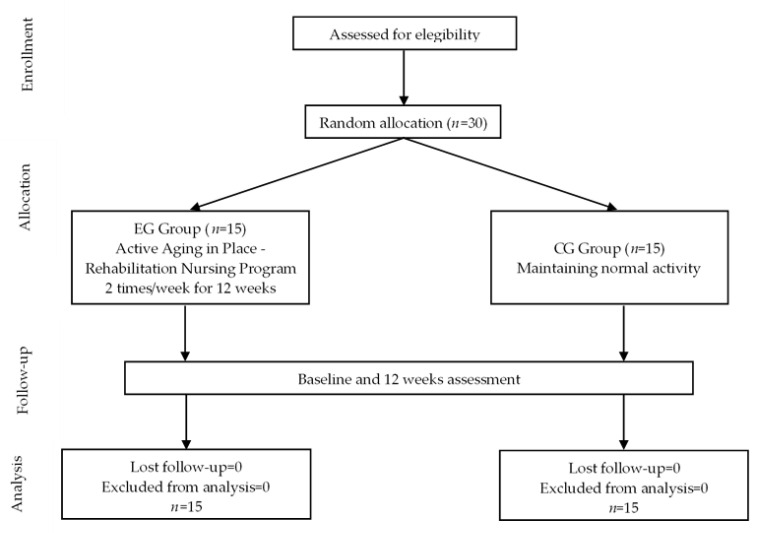
CONSORT flow diagram of participants’ flow through study.

**Table 1 healthcare-11-00276-t001:** Description of the Active Aging in Place–Rehabilitation Nursing Program (AAP-RNP).

Variables	Main Exercises	Duration	Intensity	Frequency	Progression		
					1st Month	2nd Month	3rd Month
Warm-up exercises	- Breathing and energy conservation education and training - Postural training and body awareness - Execution of joint mobility exercises (shoulder and hip circumduction)	10 min	Mild	Twice a week	- Greater respiratory control and energy conservation - Better postural balance - Higher number of series in joint mobility		
Aerobic Endurance Training	- Walking - Climbing up and down stairs	5 min	Mild (up to 4) to moderate (5–6) on a Borg rating scale of perceived exertion (0–10)	Twice a week	Longer distance and more unstable/irregular walking surface, with change of pace and direction		Climb up and down stairs, progressing according to the person’s tolerance
Strength Training	- Lying-down performance of upper limb self-mobilization, rolling, bridging, and lifting with elbow load - Sitting execution of active and resisted exercises of the upper limbs (shoulder flexion/extension and abduction/adduction, self-mobilization) and lower limbs (hip flexion/extension and abduction/adduction, hip flexion with knee flexion) - Standing (squatting)	10 min	Mild-to-moderate	Twice a week	2 sets of each exercise with 10 repetitions without load	3 sets of each exercise with 12 repetitions with load adapted to each patient (40 to 50% of 1 RM)	3 sets of each exercise with 15 repetitions with load adapted to each patient (60 to 70% of 1 RM)
Flexibility Training/ Workout	- Performing exercises such “chair sit-and-reach” and “back scratch”	5 min	Mild	Twice a week	Increase the range of motion		
Balance and coordination training	- Performing static, dynamic, and dual-task balance exercises based on the Otago program [41] - Gait training on smooth, uneven, with obstacles, uphill, or downhill - Climbing up and down stairs training	10 min	Moderate	Twice a week	- Sitting balance training - Unipedal balance with support on a stable surface–2 to 3 times (series of 5 to 10 on each leg) - Static balance in plantar-flexion, in dorsiflexion, in tandem	- Unipedal balance training without support on a stable surface - −2 to 3 times (series of 5 to 10 on each leg) - Dynamic balance gait training - Tandem gait - Plantar-flexion gait (on tiptoe) - Dorsiflexion gait (on heels) - Lateral gait - Backwards gait - Gait overcoming obstacles on the ground - Static balance training with eyes closed	- Dynamic balance training while walking on regular and uneven floor, with obstacles, uphill and downhill - Short walking distance with eyes closed - Picking an object up from the floor - Gait with “external imbalance”
ADL Training	- Simulation of functional movements that enable ADLs, such as bathing, grooming, dressing/undressing, toilet use, feeding, washing dishes, wringing out wet clothes, rolling out clothes, and cleaning windows, based on the LIFE program [42]	15 min	Moderate	Twice a week	Move on to more challenging tasks combined with strength, balance, coordination, and flexibility training, performing complex movements such as pronation/supination, cubital/radial deviation of the wrist, and fine motor skills by training thumb opposability, grip, and reach		
Relaxing and stretching exercises	- Breathing exercises - Performing joint mobility/stretching exercises	5 min	Mild	Twice a week	- Execution of active head exercises (lateral tilt and rotation) - Execution of decoupling respiratory time - Execution of active exercises of the upper limbs (flexion/extension and abduction/adduction of the shoulder synchronizing with inspiration and expiration) - Performing active trunk exercises (right/left rotation and right/left bending of the trunk) - Execution of active exercises of the lower limbs (hip flexion/extension; feet flexion/extension) - Execution of the exercise “chair sit-and-reach” and “back scratch”		
Individual Counseling	- Analysis of the difficulties and barriers that limit the habit of physical activity and social participation - Analysis of architectural barriers that hinder mobility and accessibility, and increase the risk of falls - Motivate and set goals according to the expectations of the older adult - Promote habit-formation principles to health such as self-care-physical activity and social interaction that contributes to a healthy body and mind						

Note: 1 RM (repetition maximum)–the maximum load a person can lift in a single repetition over the entire range of motion.

**Table 2 healthcare-11-00276-t002:** Sociodemographic and clinical characterization of the experimental and control groups.

Variables	EG	CG
Gender n; % Female Male	11 (73.33%) 4 (26.67%)	10 (66.67%) 5 (33.33%)
Age (Mean ± Std. Deviation)	80.07 ± 4.91	81.8 ± 5
Education (years) (Mean ± Std. Deviation)	3.13 ± 0.92	2.067 ± 1.87
Marital status n; % Married Widow(er) Single	4 (26.70%) 11 (73.30%) 0 (0%)	6 (40%) 8 (53.33%) 1(6.67%)
N.º. of household members (Mean ± Std. Deviation)	2 ± 0.535	2.87 ± 1.41
Nº of diseases (Mean ± Std. Deviation)	4.75 ± 1.49	4.13 ± 1.92
N.º. of daily medications/drugs (Mean ± Std. Deviation)	6.42 ± 1.78	6.2 ± 2.11

Note: EG-experimental group; CG-control group.

**Table 3 healthcare-11-00276-t003:** Evaluations obtained at baseline and post-program for the variables of multidimensional frailty and physical frailty.

Variables		Baseline (Mean ± Std. Deviation)	Post-Program (Mean ± Std. Deviation)	*z*	*p*
Multidimensional Frailty	Physical frailty (EG)	6.27 ± 0.88	5.00 ± 2.27	−2.39	0.017
	Physical frailty (CG)	6.0 ± 1.69	6.07 ± 1.34	−0.38	0.705
	Psychological frailty (EG)	3.53 ± 0.52	2.4 ± 1.18	−2.89	0.004
	Psychological frailty (CG)	2.87 ± 1.06	3.13 ± 0.99	−1.63	0.102
	Social frailty (EG)	1.6 ± 0.74	0.8 ± 0.78	−3.46	0.001
	Social frailty (CG)	0.93 ± 0.89	1.07 ± 0.88	−1.00	0.317
	Total frailty (EG)	11.4 ± 1.35	8.2 ± 3.43	−3.19	0.001
	Total frailty (CG)	9.8 ± 3.03	10.27 ± 2.60	−1.27	0.206
Physical frailty	Fried Frailty Phenotype Criteria (EG)	3.07 ± 0.26	2.8 ± 0.56	−2.00	0.046
	Fried Frailty Phenotype Criteria (CG)	3.133 ± 0.35	3.133 ± 0.35	0	1

Note: EG: experimental group; CG: control group; *z* and *p* values were calculated using the Wilcoxon matched pairs signed rank test.

**Table 4 healthcare-11-00276-t004:** Evaluations obtained at baseline and post-program for the variables of functional fitness, BADL, IADL, grip strength, balance, and subjective perception of effort.

Variables		Baseline (Mean ± Std. Deviation)	Post-Program (Mean ± Std. Deviation)	*z*	*p*
Functional Fitness	Chair stand test (EG)	12.27 ± 3.59	15.00 ± 3.46	−3.43	0.001
	Chair stand test (CG)	11.53 ± 2.20	10.93 ± 1.39	−1.78	0.075
	Arm-curl test (EG)	14.2 ± 2.14	17.77 ± 2.10	−3.44	<0.001
	Arm-curl test (CG)	13.53 ± 1.88	12.93 ± 1.34	−2.08	0.037
	Back scratch test (EG)	−37.73 ± 11.29	−32.27 ± 13.02	−2.79	0.005
	Back scratch test (CG)	−39.07 ± 9.63	−38.33 ± 10.27	−1.21	0.228
	Chair sit-and-reach test (EG)	−12.2 ± 7.58	−8.33 ± 7.68	−3.43	0.001
	Chair sit-and-reach test (CG)	−10.07 ± 11.51	−13.87 ± 5.69	−0.73	0.465
	Timed up-and-go test (EG)	18.833 ± 3.68	15.657 ± 4.12	−2.92	0.004
	Timed up-and-go test (CG)	19.133 ± 2.92	20.00 ± 2.92	−2.38	0.018
Grip strength	Handgrip strength (EG)	14.087 ± 4.49	17.693 ± 6.07	−3.41	<0.001
	Handgrip strength (CG)	11.867 ± 2.10	11.5 ± 2.163	−1.84	0.066
Dependence in BADL	Barthel Index (EG)	77 ± 7.973	80.00 ± 9.45	−2.71	0.007
	Barthel Index (CG)	72.67 ± 7.528	71 ± 8.062	−1.63	0.102
Dependence in IADL	Lawton Scale (EG)	9.73 ± 1.033	10 ± 1	−1.41	0.157
	Lawton Scale (CG)	11.93 ± 2.815	11.8 ± 2.859	−1.41	0.157
Balance	Single-leg balance (EG)	3.8 ± 1.26	5.2 ± 1.612	−3.31	<0.001
	Single-leg balance (CG)	3.5 ± 0.9636	3.267 ± 0.753	−1.89	0.059
	Tinetti Index (EG)	17.87 ± 2.588	18.93 ± 2.52	−2.32	0.020
	Tinetti Index (CG)	16.53 ± 4.103	16 ± 3.854	−2.12	0.034
Subjective perception of effort	Borg Scale (EG)	6.73 ± 1.033	5.53 ± 1.125	−3.63	<0.001
	Borg Scale (CG)	6.07 ± 0.704	6 ± 0.756	−1	0.317

Note: EG: experimental group; CG: control group; *z* and *p* values were calculated using the Wilcoxon matched pairs signed rank test.

**Table 5 healthcare-11-00276-t005:** Evaluations obtained at baseline and post-program in the ILP dimensions and ILP total dimensions.

Variables		Baseline (Mean ± Std. Deviation)	Post-Program (Mean ± Std. Deviation)	*z*	*p*
Dimensions of the Individual Lifestyle Profile (ILP)	Relational behavior (EG)	0.67 ± 0.724	4.53 ± 1.457	−3.45	0.001
	Relational behavior (CG)	1.6 ± 1.549	1.53 ± 1.506	−1.00	0.317
	Physical activity (EG)	0.13 ± 0.352	3.73 ± 1.4.38	−3.36	0.001
	Physical activity (CG)	1.33 ± 1.234	1.33 ± 1.234	0	1
	Stress management (EG)	5.07 ± 1.486	5.8 ± 1.320	−3.05	0.002
	Stress management (CG)	3.87 ± 2.615	3.73 ± 2.52	−1.41	0.157
	Nutrition (EG)	3.93 ± 1.100	4.27 ± 1.22	−1.67	0.096
	Nutrition (CG)	5.4 ± 2.72	5.27 ± 2.549	−3.42	0.001
	Preventive behavior (EG)	7.87 ± 1.06	7.87 ± 1.06	0.00	1
	Preventive behavior (CG)	6.4 ± 2.261	6.27 ± 2.219	−1.00	0.317
Total ILP (EG)		17.67 ± 3.13	26.8 ± 4.057	−3.41	<0.001
Total ILP (CG)		17.27 ± 5.738	18.13 ± 5.693	−1.93	0.053

Note: EG: experimental group; CG: control group; *z* and *p* values were calculated using the Wilcoxon matched pairs signed rank test.

## Data Availability

The data that support the findings of this study are available from the corresponding author upon reasonable request.

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
