# Peer review of "Effect of the Active Aging-in-Place–Rehabilitation Nursing Program: A Randomized Controlled Trial"

_healthcare, 2023, doi:10.3390/healthcare11020276_

Round 1

Reviewer 1 Report

This manuscript is a multicomponent intervention study for a frail community d-welling older people. Much has been published in recent years and there is a lot of evidence on the type of exercise, duration, cadence, etc....

It is important for the authors to emphasise what this study adds to the existing literature, and they should therefore review the recent literature on the latest meta-analyses.

It also has shortcomings in terms of the sample and the type of analysis of results.

The authors should also indicate who carried out the assessments, the order in which they were carried out, the adherence of the subjects to the sessions, whether there was a difference according to adherence, etc....

I feel that the article is well written but once they add the shortcomings indicated they should emphasise what it adds taking into account the limitations of the sample and the analysis carried out.

Minor: please, the authors should change the concept "elderly" to "older people" throughout the manuscript.

Author Response

Dear Reviewer:

Enclosed you will find a revision of our manuscript, “The Effect of Active Aging in Place – Rehabilitation Nursing Program”. We would like to thank you for their thoughtful and constructive comments. We have considered all of the suggestions and have incorporated them into the revised manuscript. Changes to the original manuscript are highlighted in yellow, and we believe that our manuscript is stronger as a result of these modifications. A response to the journal requirements and the reviewers’ comments is presented below.

Reviewer 2 Report

Dear Editor,

Thank you for the opportunity to review the article entitled The Effect of Active Aging in Place – Rehabilitation Nursing Program, with the aim of The aim of 20 this study was to evaluate the effect of Active Aging in Place - Rehabilitation Nursing Program 21 (AAP-RNP) on the functional capacity and lifestyles of frail elderly.

The article is interesting and needs some corrections:

-In the title put the Method of the study

- In the type of study it is not clear what kind of study it is, is it a quasi-experimental study, a randomized controlled trials? In some places it is referenced in one way and in others in another

-Identify in the introduction similar studies, what is the relevance of this study compared to others

- In the methodology: explain the randomization method

- Explain how the evaluation was done, if it was blinded, or if it was done by the person who did the program. If the latter option is used, it should be stated as a limitation of the study.

Author Response

(The authors gave the same response as above.)

Round 2

Reviewer 1 Report

I feel that the authors have improved the manuscript, although I feel that they should do a more thorough analysis, I leave it up to the editor to decide

Author Response

Dear Reviewer:

Thank you for contributing to the improvement of the manuscript. As proposed, the literature was reviewed and several recent reviews and meta-analyses on the topic were included to highlight the information provided by this new study.

Clarifications were also provided about the study, setting the sample (the sample size was scarce due to the restrictions of the Covid-19 pandemic), the randomization, the evaluations and the steps that were followed to carry it out, which became clearer.

We add that the Experimental Group had adherence to the 24 sessions of the program greater than 95% (not 100% due to acute illness) and we highlighted in manuscript in yellow.

It is for all these reasons that we are grateful for all your proposals, which we consider resolved and which have enriched the study with the necessary quality for its publication, so we reiterate our thanks for your contributions.

Reviewer 2 Report

Dear Editor,

The authors have significantly improved the quality of the article.

Congratulations!

Excellent work.

Author Response

Dear Reviewer:

Thank you for contributing to the improvement of the manuscript.
